# The Artificial Intelligence of Things Sensing System of Real-Time Bridge Scour Monitoring for Early Warning during Floods

**DOI:** 10.3390/s21144942

**Published:** 2021-07-20

**Authors:** Yung-Bin Lin, Fong-Zuo Lee, Kuo-Chun Chang, Jihn-Sung Lai, Shi-Wei Lo, Jyh-Horng Wu, Tzu-Kang Lin

**Affiliations:** 1National Center for Research on Earthquake Engineering, Taipei 106, Taiwan; yblin@narlabs.org.tw; 2Hydrotech Research Institute, National Taiwan University, Taipei 106, Taiwan; windleft@gmail.com; 3Department of Civil Engineering, National Taiwan University, Taipei 106, Taiwan; ciekuo@ntu.edu.tw; 4National Center for High-Performance Computing, Hsinchu 300, Taiwan; LSW@nchc.narl.org.tw (S.-W.L.); jhwu@nchc.narl.org.tw (J.-H.W.); 5Department of Civil Engineering, National Yang Ming Chiao Tung University, Hsinchu 300, Taiwan; tklin@nctu.edu.tw

**Keywords:** bridge failure, scour monitoring, flood, MEMS, deep learning

## Abstract

Scour around bridge piers remains the leading cause of bridge failure induced in flood. Floods and torrential rains erode riverbeds and damage cross-river structures, causing bridge collapse and a severe threat to property and life. Reductions in bridge-safety capacity need to be monitored during flood periods to protect the traveling public. In the present study, a scour monitoring system designed with vibration-based arrayed sensors consisting of a combination of Internet of Things (IoT) and artificial intelligence (AI) is developed and implemented to obtain real-time scour depth measurements. These vibration-based micro-electro-mechanical systems (MEMS) sensors are packaged in a waterproof stainless steel ball within a rebar cage to resist a harsh environment in floods. The floodwater-level changes around the bridge pier are performed using real-time CCTV images by the Mask R-CNN deep learning model. The scour-depth evolution is simulated using the hydrodynamic model with the selected local scour formulas and the sediment transport equation. The laboratory and field measurement results demonstrated the success of the early warning system for monitoring the real-time bridge scour-depth evolution.

## 1. Introduction

The collapse of bridges due to foundation scour is a worldwide concern to the public. Between the 1960s and 1990s in the United States, more than 1000 bridges collapsed, among which more than 60% were damaged by scouring [1]. Similar problems exist in East Asian countries, especially in areas subject to floods. For instance, in Taiwan, there are over 20,000 bridges crossing rivers. During Typhoon Morakot in 2009, an extreme amount of rainfall triggered severe flooding throughout southern Taiwan, resulting in approximately 150 bridges collapsing in this single flood event [2]. Several bridge failure cases in Typhoon Sinlaku (2008) that occurred during floods in Taiwan are shown in Figure 1. Causing reduction of structural capacity during a flood, scour failure of the bridge pier structure tends to occur suddenly and without prior warning. It is crucial to monitor real-time scour-depth variations and avoid catastrophic failure that brings about significant operational disruption or loss of life [3,4]. A real-time bridge scour monitoring system should monitor the recorded scour depth during times of flooding. Maximum scour may occur at the highest flood discharge, at which the scour hole tends to be refilled as floodwater recedes. The scour monitoring information helps assist engineers in designing safer and more cost-effective bridges.

The combined effects of complex flow vortex systems around piers involve time-dependent flow patterns, sediment transport mechanisms, and the interaction of pier structures, making the scour processes extremely complicated [3]. Many formulas for estimating scour depth at bridge piers can be found in the literature [4]. Most of the scour studies derived from experimental flume results mainly focus on the maximum scour depth by empirical regression formulas. Moreover, most of the data used in scour formulations were obtained from the laboratory rather than from the field. However, field data are limited due to observational difficulties [5]. These empirical formulas may not be accurate enough for field applications without sufficient field-measured data as inputs. Therefore, it is necessary to develop a field-based real-time monitoring system to observe and identify the scour-depth evolution at the bridge pier.

A scour hole is generally backfilled as flood flow recedes. Bridge inspections after the flood are not sufficient to fully determine the deepest extent of scouring damage at the time of the highest flood discharge. It is difficult and dangerous to acquire in situ scour-depth information by inserting a heavy object or the like into the riverbed for direct measurements in flood. Therefore, critical bridges are subject to scour, and safety monitoring is required. A real-time scour monitoring system can monitor bridge safety and avoid unnecessary maintenance. Nevertheless, the development of measuring instruments with data acquisition systems generally faces difficulties in surviving, particularly the bumping of floating debris or large-sized sediments during flood events (Figure 2). Therefore, reliability and robustness are important design concerns for effective and long-term monitoring in the system, especially when installed sensors must be buried deep in the riverbed filled with water under pressure. Under the impact of rapid floods and drifting objects, the scour monitoring system may be vulnerable if it is not adequately protected. In addition, the complicated turbulent flow with floating debris and sediments surrounding the bridge pier may also affect the reception and accuracy of signals.

Recently, researchers have paid more attention to bridge-scour monitoring technology development. Most approaches directly attach the sensory system to the pier or embed sensors in the riverbed. When the sensor is exposed from the riverbed, the scour depth is then detected through the corresponding signal from the sensor. From the literature, the most used approaches for scour monitoring or counter measurement include methods such as fiber optics sensors [6,7,8,9,10,11,12,13,14,15], micro-electro-mechanical systems (MEMS) sensors [16], acoustic emission (AE) sensors [17], the vibration-based method [18,19,20,21], the sonic echo approach [22], time-domain reflectometry (TDR) [23,24,25,26,27,28,29], piezoelectric sensors [30,31], temperature sensors [32], smart rocks [33], image sensors [34], strain sensors [35], and bridge/pier nature frequency variation [36,37,38,39,40,41]. Prendergast and Gavin (2014) reviewed and reported a broad range of instrumentation techniques for scour-depth measurement [38]. The scour monitoring is sensitive to environmental conditions, such as temperature, salinity, turbidity, air, sediments, and debris. For example, both sonar and radar can receive large amounts of noise caused by flow turbidity and suspended granular particles, making these systems unreliable for real-time monitoring of the scour progress. The electrical resistance effect has been widely used, such as time-domain reflectometry (TDR) and piezoelectric sensors for scour detection and vortex-induced vibration monitoring to observe its changes in the permittivity of the medium around bridge foundations. TDR works by generating electromagnetic pulses that can attenuate and disperse their signals due to the transmission line or the magnetic conduction problems of the monitoring environment. This shortcoming of TDR reduces its ability to detect subtle changes in the scour process. A vibration-based sensor for scour measurement was developed recently, which may have a greater potential for environmental and operational sensitivities than TDR and sonar devices. However, an extreme flow condition with heavy debris impact would result in bridge failure in flood. Instrument development with data acquisition systems usually faces challenges in surviving the flood event. Hence, the vibration-based arrayed sensors proposed in the present study are housed in stainless steel balls and provided adequate protection by filling the ball with resin for long-term waterproofing performance as they are buried in the riverbed.

Wireless sensor network (WSN) systems with the Internet of Things (IoT) and artificial intelligence (AI) technologies used as promising technologies in civil engineering, smart city, and industrial applications in recent years have greatly improved the approaches of structural health monitoring. The Internet of Things and big data analytics have been proposed and applied for many applications. They provide spatial data over large areas over time and are suitable for many monitoring applications. Intelligent and wireless networking sensors collect and process large amounts of data, including monitoring and analyzing structural damage, traffic conditions, and weather and flood conditions, etc. In the present study, a monitoring system that consists of a combination of IoT and AI is developed and utilized to obtain real-time measurements in the scour processes at the bridge pier. In recent years, with the rise of deep learning (DL) technology, especially in image recognition and classification applications, scientific fields have made impressive breakthroughs [42,43,44,45,46,47]. The significant difference between convolutional neural networks of deep learning and conventional machine learning (ML) methods is that the traditional ML methods rely on humans to design and select the image features for classification where the corresponding results are deeply affected. In particular, image features designed are often straightforward and limited to human judgment, which is difficult for complex and changeable image applications. In contrast, the convolutional neural network (CNN) covers the image features generated by DL in its network architecture and generates many specific features, which suit all types of diverse image scales through a magnificent amount of image training. In general, machine learning requires engineers to define the model features and classify the data in advance, while deep learning alternately uses neural networks to discover new models and continue improving themselves. These trained specific features and characteristics summarized from actual life photos are far more versatile than those designed by human experiences. On the other hand, the multi-layer structure in the training–recognition network also contains the complexity contents for image description. Moreover, image recognition and classification problems are the most successful applications of deep learning. In the present study, the water levels as time-series variables during flood events are critical for hydrological measurement, analyzed by image recognition through deep learning technologies [48,49,50].

Although many types of research focused on bridge scour have been reported in the literature, numerical simulations of scour-depth variations with in situ measurements during floods are limited. Proper prediction of temporal scour-depth changes by the numerical hydrodynamic model can deliver helpful early warning information for bridge failure. To predict riverbed changes is vital to recognizing any possible aggravation and degradation of the riverbed level in response to river morphology, typically around the bridge piers. Regarding the phenomena of a mobile riverbed in floods, sediment transport can simulate the flow field and pattern by a numerical hydrodynamic model to assess the time variation of the total scour depth of the pier. The sedimentation and river hydraulics two-dimensional (SRH-2D) model is a depth-averaged 2D hydraulic and sediment model, simulating flows in fluvial rivers with hydraulic structures such as bridge piers [51]. The model is suitable for modeling riverbed migration, such as the general scour of the riverbed. However, the numerical hydrodynamic model cannot compute the local scour depth directly. With commonly used empirical formulas, the local scour depth is calculated using the simulated outcomes of water level and velocity from the SRH-2D model. For practical engineering purposes, the total scour depth is the sum of the general scour from the numerical hydrodynamic model and the local scour calculated by the empirical formulas. The predictive information of scour-depth variations can provide valuable data for early warning in bridge failure.

In this article, the primary purpose is to develop a scour monitoring system for early warning in highway bridge failure that includes field-monitoring sensors and scour-depth simulation. By adopting IoT sensing and wireless communication technology, the system is proposed and designed with vibration-based arrayed MEMS sensors that are implemented in situ to obtain real-time scour-depth measurements of scour and deposition processes. The sensors are conceptually demonstrated in the laboratory flume before they are installed on the study site. The real-time floodwater-level changes around the bridge pier are performed using CCTV images by the Mask R-CNN deep learning model. The total scour-depth evolution is simulated using the hydrodynamic model with the selected local scour formulas and the sediment transport equation. The predictive information of scour-depth variations can provide valuable data for early warning in bridge failure.

## 2. Scour Monitoring System

### 2.1. Laboratory Conceptual Validations

The experimental test is to validate the newly designed bridge-scour sensory device consisting of serial arrayed cantilever beams with MEMS accelerometers. The cantilever beam sensing principle was widely used to measure the vibration of MEMS sensors. The proposed monitoring system was verified to evaluate the responses of bed-level variations in a laboratory flume. The experiments were conducted in a 12 m-long, 0.3 m-wide, and 0.6 m-deep flume with glass sidewalls at the Hydrotech Research Institute of National Taiwan University, Taipei, Taiwan. As shown in Figure 3a, the bridge pier with the arrayed cantilevered beams is placed in the middle of the sand paved flume. The prescribed discharge and its corresponding water depth were controlled by adjusting the inlet valve and tailgate. Before the test began, all the sensors were buried by uniform sand of 0.9 mm in diameter and then slightly compacted by a plate hammer. The experiment was carried out at a steady flow rate with a fixed water level.

At the beginning stage shown in Figure 3b, water flowing from the inlet gradually seeped into the paved layer, which became saturated and led to a disturbance of the paved layer, thus causing the signals of Sensors S1 and S2 to be larger than those of Sensors S3 and S4. From the signal of Sensor S1, it was found that scour started at time t1 because the sand bed was eroded around Sensor S1, and the cantilever beam was gradually exposed to the flowing water. The fluctuated signal obtained from Sensor S1 combined the dynamic drag force and flowing water turbulence around the bridge pier. Due to dynamic flowing water impact, the cantilever beam with Sensor S1 vibrated at a slightly backward position. More significant vibration signal fluctuations were also observed because of coarse sands hitting the cantilever beam. The signal recovered a smaller vibration signal due to fewer suspended particles with an equilibrium state scour progress at time t2 subject to the steady water drag force. In order to observe the cantilever beam signal of the sensing system in the deposition process, sand was added at the upstream side of the pier at t3. As Sensor S1 was covered by sand to diminish dynamic drag force, the cantilever beam gradually reverted to its original position. This deposition process was observed from the voltage of the decreased vibrated signal and revealed that this cantilever beam is then buried in the bed. The unexpected scouring signals observed at t4 were suddenly adding some sand in front of the pier. At time t5, scouring occurred again. As mentioned above, the signal caused a sharp reciprocating signal due to the suspended particles acting on the cantilever beam, acting by the flowing water at t6, similar to t2. The tailgate was opened at t7 to drain flume water. This experimental test was conducted in a steady-state flow rate, in which the velocity and the discharge of flowing water are insufficient to wash out Sensors S3 and S4. Evidently, the steady-state flow scour experiment verified that the conceptually designed cantilever beam sensing system can be used to measure the scour and deposition processes.

### 2.2. Vibration-Based MEMS Sensor

Measuring vibration is critical for detecting and diagnosing any deviation from normal situations. Vibration measurements by the use of conventional accelerometers are well known and accepted. Recent advances in IoT system technologies with MEMS sensors offer great promise for the future of smart monitoring. MEMS accelerometers are one of the simplest and most suitable MEMS systems, which are indeed a mature sensing technology. The technology and applications behind MEMS accelerometers have been widely investigated, with MEMS accelerometers being sensors that detect linear acceleration.

Due to the widespread adoption of MEMS accelerometers in smartphone devices, there are many types of accelerometers available in the market today. In this paper, the vibration-based sensor of LSM303DLH, STMicroelectronics is used as a scour positioning indicator. The LSM303DLH is a digital 3-axis accelerometer. As shown in Figure 4, the LSM303DLH accelerometer sensor is housed in a stainless steel ball with Recommended-Standard-485 (RS-485) signal line and protected in a full-fill resin that isolates short-circuits or signal loss caused by flood and debris impacts. Appropriate packaging to resist harsh environments was the main long-term robustness issue for bridge-scour measurements. Herein, these enhanced scour-sensor assemblies in a rebar cage were used for measuring the vibration signals as they emerged out of the riverbed and vibrated due to the flowing water. These packaged LSM303DLH accelerometers for bridge-scour measurement were tested with different flow velocities in the laboratory to ensure their performance, particularly the long-term waterproof characteristics and the WSN function, before they were deployed in the field.

## 3. Field Deployment and Results

### 3.1. Site Description and System Installation

The Da-Chia Bridge on National Highway No. 3 in central Taiwan spans the Da-Chia River between two major areas (Figure 5): Miaoli County (northbound) and Taichung County (southbound). As a major transportation route with heavy traffic, the 1.2 km-long bridge with 38 spans was constructed in 2003 using cylindrical columns as piers (2.7 m in diameter). It is located 6.57 km away from the estuary of the Da-Chia River, flowing westward into the Taiwan Strait. According to hydrological data, the natural drainage area of the upper reaches of the Da-Chia River is 1981 square kilometers. The bed slope of the river reach around the bridge is about 1%. According to the data provided by the Water Resources Agency, Taiwan, the peak discharge of the 100-year return-period flood as the design flood for levee protection is estimated as 21,000 m^3^s^−1^ based on the frequency analysis. The mean annual discharge at the Da-Chia Bridge is around 116 m^3^s^−1^. The mainstream flows underneath the Da-Chia Bridge between pier P26 and pier P28. The scour monitoring system was installed on the bridge pier P28, as shown in Figure 6. With five vibration-based MEMS sensors, the scour monitoring system was designed for measuring the bridge scour-depth variations. Packaging the sensors in a waterproof stainless steel ball to embed into the riverbed can be proved to resist a harsh environment in floods for long-term operational purposes. These sensors were tested in the laboratory for durability and reliability before being deployed in the field. Moreover, the wireless station contained the wireless sensor network (WSN) systems and the CCTV camera with a night infrared light source focusing on the water flow around the pier. Hence, the system design is robust and efficient to withstand debris impact, sustain long-term service, and survive flooding events.

In the present study case, the flood event occurred on 19 May 2014 with heavy rainfall generated in the watershed. The study domain for the numerical hydrodynamic modeling is located between two surveyed cross-sections imposed as the upstream inflow boundary at 3285m away from the bridge and the downstream boundary condition (BC) at the railroad bridge shown in Figure 7. The bed elevations in the simulation domain ranged from EL. 51.89 m to EL. 97.50 m above sea level. The median size of riverbed sediments was 91.5 mm. The upstream inflow discharge was provided by the Shigang Reservoir Management Center, Taiwan. The flood hydrographs plotted in Figure 7 have multiple peak discharges and water levels with three flood peaks of 839 m^3^s^−1^, 885 m^3^s^−1^, and 1011 m^3^s^−1^, sequentially. The water level started rising on 19 May, and approximately after 2 days, the highest flood peak arrived at the bridge on 21 May. The monitoring data of scour-depth evolution at P28 were adopted for numerical modeling verification. The main objective of the present study was to evaluate the applicability of scour-depth simulation for developing an early warning system at the Da-Chia Bridge on National Highway No. 3.

### 3.2. Mask R-CNN Object Detections and Recognitions

Due to the complexity of the floodwater flow, the water levels as time-series variables are critical for hydrological measurement, which can be analyzed by image recognition through deep learning technologies. For recognizing objects in images, computer vision technologies have a wide range of human portrait recognition, object detection, and human posture prediction derived from object recognition. Typically, visual sensors such as CCTV are employed for non-contact, remote telemetry, and long-term monitoring [52], e.g., water-level monitoring [53]. The training models of image automatic identification to the water level around the bridge piers are carried out herein. The corresponding deep learning models are then used for image verification and water level recognition. As well known, following the deep learning in Convolutional Neural Network (CNN) development [54,55], another practical Region-based Convolutional Neural Network (R-CNN) has been applied to recognizing and detecting multiple objects [56]. Mask R-CNN is one of the Region-based CNN family (R-CNN) [56], Fast R-CNN [57], and Faster R-CNN [58], which has the highest accuracy in the Microsoft Common Objects in Context (COCO) segmentation challenge, and it is being used extensively for different instance segmentation applications [59]. Microsoft COCO is a large-scale dataset that aims to enable future research for object detection, instance segmentation, image captioning, and person key points localization. It is usually used as the performance benchmark for model evaluation [60]. The object detection approaches take advantage of CNN-like technologies coupling with multi-region in multi-resolution parallel recognition capabilities and are popularly used image object detection methods [61]. These algorithms have been applied to many machine vision-related fields, such as the visual recognition of robots and autopilot cars, the defect recognition of online products, and automatic recognition of security systems, etc. [62,63,64]. Mask R-CNN is an extension of Faster R-CNN, which detects the mask at the pixel level for each object detected rather than the bounding box. This method has advantages when calculating pixel-level contours, such as the boundaries of water bodies and bridges. Previous research has shown that using the segmentation network to detect floods and calculate the flood depth is a practical approach [65,66,67].

The Mask R-CNN is adopted herein to identify and analyze images taken from the CCTV camera, focusing on the water surface around the pier to map the water level due to the complexity of the flood. The architecture framework of segmentation shown in Figure 8 is mainly based on the Faster R-CNN expanding method for object detection, and the Mask R-CNN is retaining the RPN (Region Proposal Network) layer of the original architecture. For the practice of transfer learning, the Mask R-CNN is implemented by using Keras with a TensorFlow backend and based on pre-trained weights of the COCO dataset [66]. The convolutional backbone architecture (Conv in Figure 8) is the ResNet-101 [60] employed for feature extraction over an entire image. The image pixels in CNN are essential and are used in classification algorithms to classify or segment the regions of interest (ROI). The purpose of this pixel-level classification algorithm is used to pool the ROI into the Faster R-CNN. The ROI-Align algorithm also replaces the algorithm to generate a fixed-size feature map. The instance segmentation is used to classify the background/foreground regions and FC (Fully Connected layer) objects and the difference of bounding boxes. These methods are used as semantic segmentation by adopting FCN (Fully Convolutional Network) as a pixel-level classifier, combining the classification results of the two parties as to the output. The so-called two-stage segmentation uses the bounding box of FC to narrow down the candidate area of FCN. The advantage of the segmentation network is that both flooded and non-flooded instances can be used to train the network to learn the features of the pier. Generally, the presence of non-flooded images is more frequent and accessible to collect rather than flooded images. Moreover, this method can be applied to estimate the water level based on the reference object with a known height. The output images are converted from the perspective transformation and map the water level (white dash line) by normalizing the scale in the range [0,1]. Then, the actual water level based on the exposed bridge piers is estimated.

The CCTV images during flood events were recorded every minute in real-time monitoring and selected in the study case during flood at pier P28 by daily images. The images displayed in Figure 9a for the demonstration were obtained from 19 May to 24 May 2014 with the recorded time on the top, each image representing the various floodwater-level situations. The Mask R-CNN deep learning model automatically performs the turbulent rush water around the bridge pier from real-time video images. Figure 9b shows that the flowing water is marked in cyan, while the pile cap or stones on the river bed are marked in red. It is noted that the foundation of the pier is not displayed to highlight the range of the rush flood flow by recognizing the rising or falling edge of the water surface.

The floodwater surface images were recorded and transmitted through the wireless station with a CCTV camera at pier P28 in real-time monitoring. The layout of the scour monitoring system is illustrated in Figure 10a, which indicates the setup locations of scour sensors and the target region of the image taken. Figure 10b shows that the bridge pier is marked in green, and the flowing water is marked in cyan. Each image obtained is transformed to make the pier straight and normalize scale in the range [0,1] as a water level ruler. The amount of these green pixels reflects the water level along the pier. In Figure 10b, at time t, a white dash line is recognized and marked to identify the edge of the water surface on the normalized figure of the pier, and similarly, the water surface edge marked by a white dash line is also recognized as shown in Figure 10c at time t+∆t to reveal water level rising. Hence, the water levels can be obtained from the scale multiplied by the height of the pier automatically. Based on the Mask R-CNN deep learning model framework, the recognized water levels in hourly time series are plotted in Figure 11, which are important as the field measurements to provide data for hydrodynamic model calibration and validation. The image recognition data validate the simulated water levels obtained from the hydrodynamic model. The water level and velocity hydrographs are synchronized with each other.

### 3.3. Scour Monitoring Results and Analysis

The scour monitoring system was installed on bridge pier P28 in the mainstream of the Da-Chia River (Figure 6). All the systems, including the wireless station, were fabricated in the laboratory. The scour monitoring system was subject to floating debris impact. Before being used in the field, all the sensors with waterproof package were tested and analyzed for long-term durability and performance.

The sensor signal data were collected at P28 from 19 May through 23 May in 2014, as shown in Figure 12. To assess the safety of the bridge foundation, the duration of the flood event is herein separated into three periods: T1 is the time before scouring, T2 presents the vital scour/deposition processes, and T3 is the period of flood recession. The zero initial scour-depth level set to its corresponding level of the riverbed at P28 was EL.70.90 m, at which Sensor S1 was placed. The five vibration-based sensors were denoted as Sensor S1 (position at EL.70.90 m) through Sensor S5 (at EL.68.90 m). The distance between each sensor was 0.5 m. Each sensor was packaged within a waterproof stainless steel ball used as a scour positioning indicator. They were protected in a steel reinforcement cage and deployed on the side of the pile cap at the pier foundation (Figure 6). Once the scour monitoring sensor emerges from the riverbed, it sends vibration signals as it vibrates due to the turbulent flow. In Figure 12, Sensors S1 and S2 are gradually eroded out after 10:00 on 19 May as the water level starts rising, and then they detect flow turbulence to reveal signal fluctuations. The strong signal fluctuations from Sensors S1 and S2 were observed after the T1 period, which is induced by the fast-rising water level in flood. At the early stage of the T2 period, around 14:00 on 19 May, the signal readings from Sensor S3 were found to receive minor vibration, presenting an evident erosive impact reaching S3. The scour depth at this moment was estimated at 1.0 m. At 19:00 on 19 May, the signal readings from Sensor S3 displayed strong vibration by powerful erosive impact at the first flood peak discharge (Figure 12). Substantial fluctuations in reading due to turbulence around the pier indicate a strong interaction between sensor and water current. If the output voltage value is larger than 1.2 or less than 0.8, it is assumed that the scouring force may extend down to a depth of 0.25 m, presenting stronger vibration signals. Thus, the scouring process reached S3, while the scour depth was estimated to be 1.25 m due to more vital readings delivered by Sensor S3. At the same time, the signals received by Sensor S2 showed relatively weak fluctuations compared with Sensors S1 and S3, which perhaps has a little stone jammed inside the rebar cage that reduces the freedom of the stainless steel ball.

After the first peak, the flood flow receded and the readings showed weak fluctuations. The water level and velocity were drawn down to the locally lowest point at 10:00 on 20 May (Figure 11). The signal readings were received only from Sensors S1 and S2; the scour depth was estimated to be 0.50 m. They presented almost no fluctuation after this moment, which may indicate that they were buried in the riverbed again due to sediment deposition. After the second peak of flood discharge arrived at 21:00 on 20 May, Sensors S1–S3 were active again and reached the scour depth of 1.00 m. When the water level reached the third peak at 16:00 on 21 May, Sensors S1–S3 continued to have significant readings as they sensed the powerful erosive impact by running flow and sediments. Sensor S4 detected a minor signal at 19:20 on 21 May, which means the flood eroded the riverbed down to the S4 position, following the highest peak water level or velocity. Sensors S1–S4 were all active to generate the maximum scour depth of 1.50 m. However, the erosive force on Sensor S4 only lasted for a short period (about 1–2 h), and then it was buried again. After the third peak, the flood started to recede, and Sensors S1–S3 kept sensing apparent water-sediment impacts until 10:00 on 22 May. Afterwards, Sensor S3 was buried under the riverbed due to sediment deposition, while the flood kept receding. Meanwhile, Sensors S1 and S2 still detected the weaker signal responses due to the low flow velocity, which should be exposed in the flowing water to remain with a scour depth of 0.50 m. Therefore, the maximum scour occurred at EL. 69.40 after the third flood peak discharge at 19:20 on 21 May. Then, the scour hole was backfilled to EL. 70.40 at 10:00 on 22 May, i.e., the riverbed dropped 1.00 m from the original elevation after this flood event. The maximum scour depth at P28 was confirmed by field investigation after the flood, as shown by the photos in Figure 6. After the T2 period, the continuous flat signal readings indicate that sediment deposits should have buried the Sensors S3–S5 in the T3 period (Figure 12). The real-time scour-depth data (with triangle symbol) obtained from the scour monitoring system are plotted against periods from T1 to T3 in Figure 13.

## 4. Scour Depth Simulation for Early Warning

### 4.1. Hydrodynamic Simulation

Bridge scour around a pier occurs mostly during high water level and velocity. The scour hole around the pier may be backfilled as the peak flow recedes. Proper prediction of temporal scour-depth variations by the numerical hydrodynamic model can provide helpful information of early warning for bridge scour failure. Practically, the total scour depth at the pier is obtained by the sum of the general scour, contraction scour, and local scour. While each span between two bridge piers along the Da-Chia Bridge is substantially longer than that of the pier diameters, the contraction scour could be neglected [68]. Herein, the total scour depth of the bridge pier is primarily dominated by general scour and local scour. Considering the bed scour of non-equilibrium sediment migration in floods, a depth-averaged two-dimensional (2D) hydrodynamic model can be used to estimate the water depth and flow velocity to assess the time variation of the total scour depth of the pier. The sedimentation and river hydraulics two-dimensional (SRH-2D) model developed by the U.S. Bureau of Reclamation is a depth-averaged 2D hydraulic and sediment model that solves 2D shallow water flow equations through unstructured hybrid meshes [51]. The model can simulate the hydrodynamic flows in fluvial rivers with hydraulic structures such as bridge piers. Moreover, the model has been tested in several experimental and field cases for modeling riverbed variations in shallow-water flows. It has been demonstrated that it is suitable for modeling riverbed aggradation and degradation, such as the general scour of the riverbed. However, the model cannot be used to compute the local scour evolution directly. Therefore, four commonly used empirical formulas are adopted to calculate the local scour depth based on the simulated hydraulic outcomes of water level and velocity from the SRH-2D model. The total scour depth is merely the arithmetic sum of the general scour from the numerical model result and the local scour calculated by the empirical formulas.

Local scour depth at the pier can be calculated by the empirical formulas proposed by Laursen (1958) [69], Shen et al., (1969) [70], Jain and Fischer(1980) [71], and Froehlich (1988) [72], etc. These commonly used formulas for local scour-depth calculation were derived under scour equilibrium conditions. The approaching flow parameters of water depth and velocity are needed and dependent on the discharge and water level hydrographs. During the flood, the time-dependent local scour-depth calculation is related to the approaching hydrodynamic parameters such as water depth, velocity, Froude number, and the geometrical parameters such as bridge noise shape, flow direction, sediment size, and pier diameter. Among these hydrodynamic parameters, water depth, velocity, flow direction, and Froude number can be directly obtained or calculated using the SRH-2D model. These commonly used local scour empirical formulas can calculate the local scour-depth evolution caused by the change of flow depth and velocity during the flood event. For practical purposes, one may select a proper formula for a specific bridge pier through a validation process by comparing the data measured by the scour monitoring system.

The flood event occurred on 19 May 2014 with heavy rainfall generated in the watershed. The computational domain for the flow simulation has 6870 computational cells and is approximately 3285 m long and 1165 m wide between two surveyed cross-sections that are imposed as the upstream inflow and downstream outflow boundaries, described in Section 3.1. As shown in Figure 7, this flood event presents multiple-peak discharge and water level hydrographs. On 21 May, the highest peak discharge arrived at the bridge. In Figure 11, the hydrographs of the simulated water level and velocity at P28 are plotted in the flood event.

### 4.2. Total Scour-Depth Evolution

As described in the previous section, the SRH-2D model is adopted to simulate hydrodynamic conditions for the approaching flow parameters to calculate local scour depth. At the same time, general scour is also calculated directly using the sediment transport equation for riverbed degradation and aggradation in the model. The temporal variations of total scour depths are calculated by combining the local scour and general scour depths to provide helpful early warning information in bridge scour failure.

General scour results from sediment material transportation on the riverbed. Based on the numerical model calibration and verification, the general scour depth was calculated using the bedload sediment transport equation by Parker (1990) [73]. According to the sediment size at the study site, Parker’s sediment transport equation is well suited for rivers composed of coarse sediments and fine sediments. The general scour depth at the pier front was simulated by the SRH-2D model, considering riverbed aggregation and degradation. For local scour-depth calculations at P28, the simulated water level and velocity varying with time, as presented in Figure 11, were used as the approaching flow parameters. The temporal variations of local scour depths from the four adopted local scour formulas were estimated and combined with the simulated general scour depths. Generally speaking, the general scour depths were relatively minor in comparison with the local scour depths during the scour and deposition processes in this flood event. The total scour-depth simulated results against time with various local scour formulas are plotted against time in Figure 13. The real-time scour-depth data obtained from the scour monitoring system were used to verify numerical modeling.

Based on the simulated results, the total scour depth combines the general scour depth and the local scour depth from 19 May to 23 May. In this flood event, starting from the original riverbed elevation, total scour depth increased immediately as the flow discharge increased while the flow velocity increased and the water level rose. Applying different local scour formulas, the trend of simulated results coincided with the measured data. However, the magnitude of the total scour depths did not fit the measured ones well for all employed formulas. Among the local scour formulas, the Froehlich formula was modified by replacing its coefficient of 0.32 with 0.50, which can obtain the best trend to fit the measured scour-depth evolution, as shown in Figure 13. Hence, the modified Froehlich formula was selected for the calculation of local scour depth in the simulation. For the first flood peak discharge at 19:00 on 19 May, the simulated scour depth was somehow overestimated, but the simulated result of the deposition process fit quite well at 10:00 on 20 May before the second flood rising. After the water level reached the third peak discharge at 16:00 on 21 May, the simulated maximum scour depth was very close to the measured data at 19:20 on 21 May, which means the flood eroded the riverbed to create the deepest scour depth of 1.50 m. Although the simulation of total scour-depth evolution overestimated in the T3 period of flood recession, the overall performance was satisfactory. The procedure mentioned above to calculate the total scour-depth evolution can provide predictive information for the early warning system.

## 5. Conclusions

Scouring at a pier remains a major cause of bridge failure. Without prior warning, scour failure at the bridge pier structure occurs suddenly and results in public transport disruption and loss of life, particularly in a flood event. Therefore, it is crucial to monitor in situ scour-depth evolution to prevent catastrophic bridge failure. A real-time bridge scour monitoring system is proposed and conceptually demonstrated in the laboratory flume in this study. The waterproof packaged monitoring technology is more flexible for harsh environments in flood events and suitable to measure the scour-depth variations during the scour and deposition processes on the riverbed.

The scour monitoring system consisting of five vibration-based sensors was installed on pier P28 of the National Highway No. 3 in the Da-Chia River, Taiwan. The system was subject to severe flooding environments. Each sensor was tested and analyzed for long-term durability before being used in the field, packaged within a waterproof stainless steel ball, and protected in a steel-bar-reinforced cage. The system was applied and evaluated in the flood event on 19 May 2014, with heavy rainfall. This flood event had multiple-peak discharge and water-level hydrographs. The real-time CCTV images of turbulent floodwater surface around the bridge pier were automatically recognized and performed by the Mask R-CNN deep learning model to estimate water-level variations. The recognized water levels in time series are essential for the field measurements to provide hydrodynamic model calibration and validation data.

The scour monitoring data in the scour and deposition processes were recorded and analyzed. As a scour positioning indicator of the scour monitoring system, the sensor could detect flow turbulence and summit signal fluctuations while it was exposed to the running floodwater. A maximum scour depth of 1.5 m was recorded from Sensor S4 vibration readings after the highest flood peak discharge regarding the original riverbed elevation. Afterward, in the period of flood recession, the continuous flat signal readings indicated that the sensors should have been buried again by sediment deposits. The maximum scour occurred at EL. 69.40 after the highest flood peak discharge; the scour hole was backfilled to EL. 70.40 afterward. This indicates that the riverbed dropped 1.00 m from the original elevation after this flood event. This maximum scour depth at P28 was confirmed by field survey investigation after the flood. Thus, the in situ measurements of total scour-depth evolution have shown quite reasonable results for the scour and deposition processes.

For early warning, the applicability of scour-depth calculation was evaluated by numerical hydrodynamic simulation at the Da-Chia Bridge. The in situ scour-depth data measured by the scour monitoring system was adopted for numerical model verification. In this study, the total scour depth of the bridge pier was mainly dominated by general scour and local scour. By adopting the SRH-2D hydrodynamic model, the general scour was calculated directly by the sediment transport equation for riverbed degradation and aggradation. Applying different local scour formulas, the trend of simulated results coincided with the measured data. Among the local scour formulas, the modified Froehlich formula resulted in the best trend to fit the measured scour-depth evolution, which was selected to calculate local scour depth in the model. While the water level reached the highest peak flood discharge, the simulated scour depth was very close to the maximum scour depth of 1.50 m. Based on the procedure mentioned previously, the overall performance of the hydrodynamic modeling for total scour-depth evolution is satisfactory. It can provide predictive information of scour-depth variations for early warning in bridge failure.

## Figures and Tables

**Figure 1 sensors-21-04942-f001:**
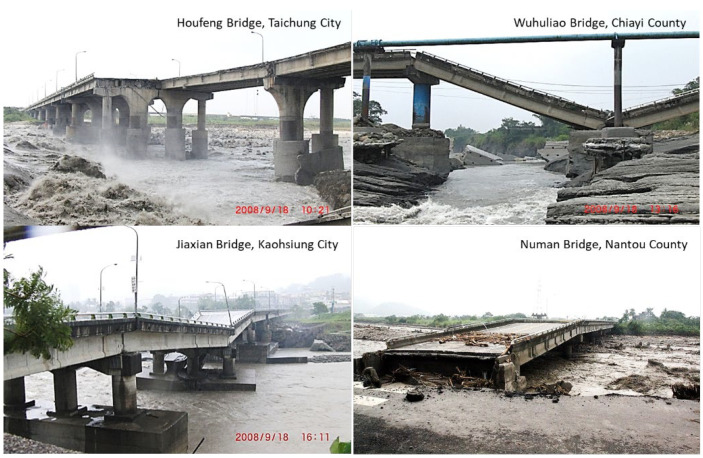
Bridge failure cases occurred during floods in Typhoon Sinlaku (2008).

**Figure 2 sensors-21-04942-f002:**
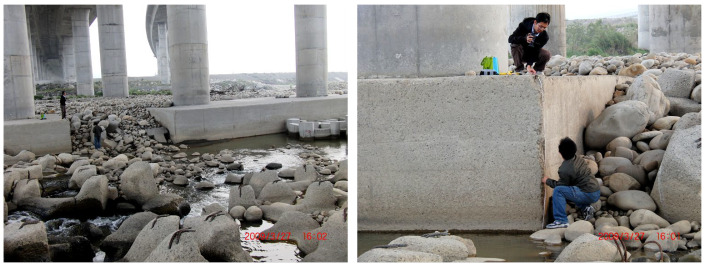
Photos at Da-Chia Bridge on National Highway No. 3 facing harsh environments with large-sized sediments.

**Figure 3 sensors-21-04942-f003:**
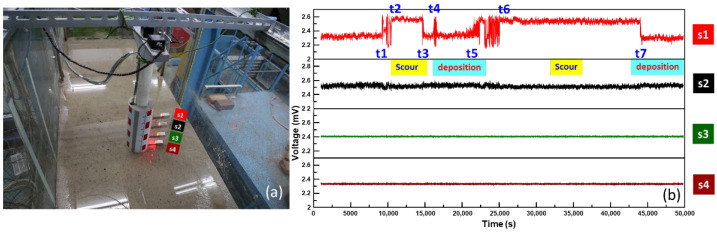
The experimental test: (**a**) the serial arrayed cantilever beams with MEMS accelerometers in laboratory flume; (**b**) signal outputs for validation of scour and deposition processes.

**Figure 4 sensors-21-04942-f004:**
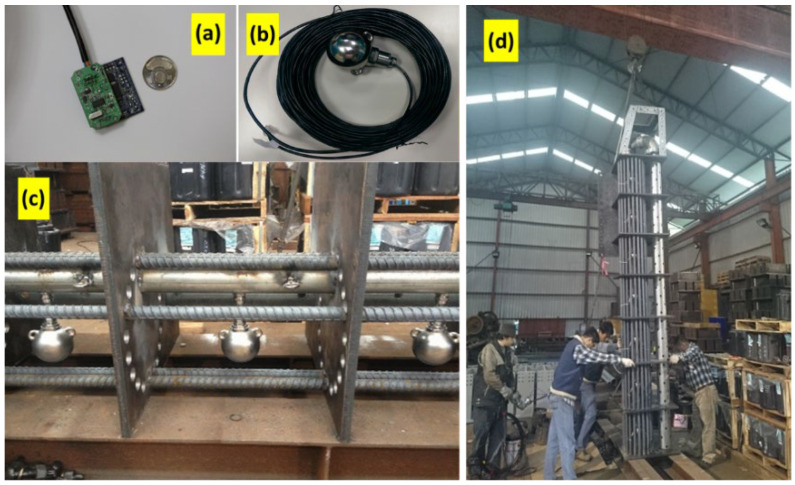
The vibration-based MEMS scour monitoring system: (**a**) sensor chip with integrated circuit (IC) and printed circuit board (PCB); (**b**) packaged scour sensors with stainless steel ball; (**c**) arrayed sensor alignment in a rebar cage at 0.5m interval; (**d**) a finished scour monitoring system.

**Figure 5 sensors-21-04942-f005:**
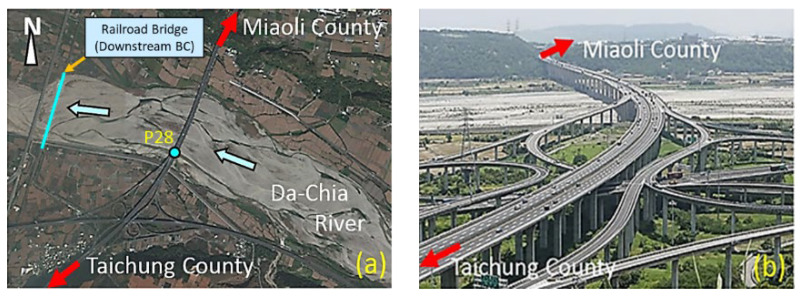
The Da-Chia Bridge spans two adjacent counties: (**a**) aerial image; (**b**) overlook photo.

**Figure 6 sensors-21-04942-f006:**
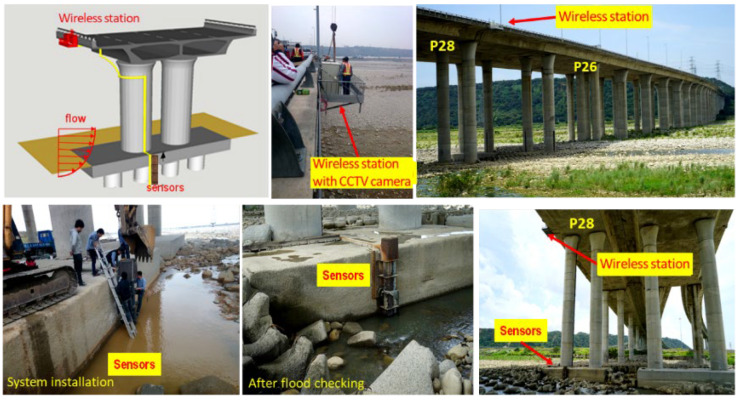
Installation of the scour monitoring system and the wireless station at P28 of the Da-Chia Bridge.

**Figure 7 sensors-21-04942-f007:**
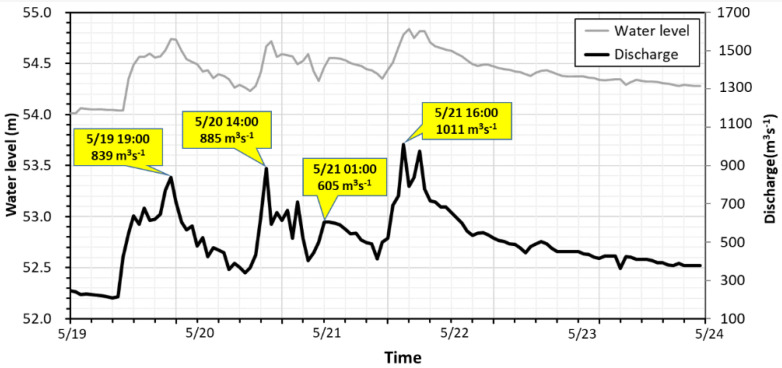
The multiple-peak discharge and water level hydrographs in the 2014 flood event.

**Figure 8 sensors-21-04942-f008:**
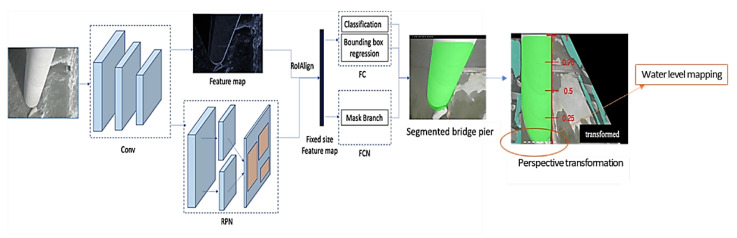
The Mask R-CNN framework of segmentation and water level estimation.

**Figure 9 sensors-21-04942-f009:**
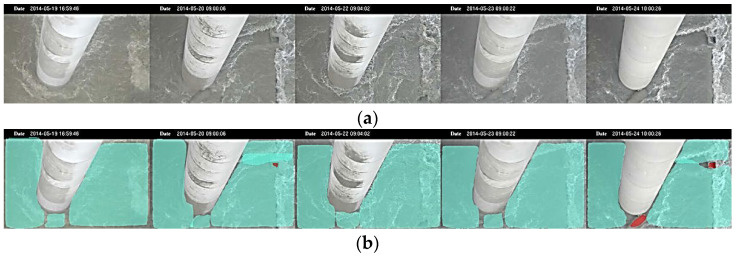
The image processes during the flood period at pier P28 with the recorded time on the top, each image representing the various flood situations at that time: (**a**) selected CCTV camera images from the real-time monitoring data; (**b**) water body segmentation results of (**a**) (cyan mask represents water, and red mask represents the pile cap on the pier or the stones on the riverbed).

**Figure 10 sensors-21-04942-f010:**
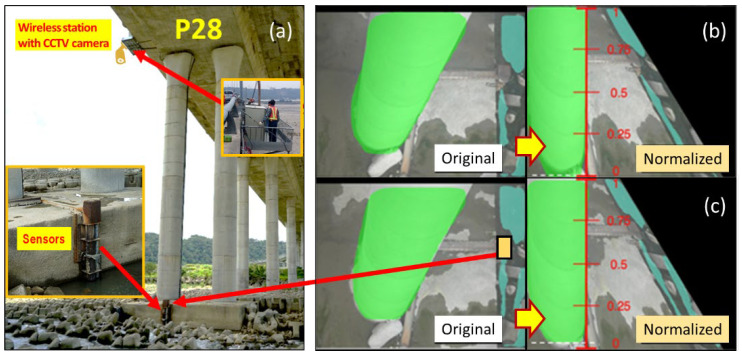
The layout of scour monitoring system corresponding to image recognitions: (**a**) installation locations of the scour sensors and the wireless station with CCTV camera at pier P28; (**b**) segmentation results of water level at time t; (**c**) water level rising at time t+∆t.

**Figure 11 sensors-21-04942-f011:**
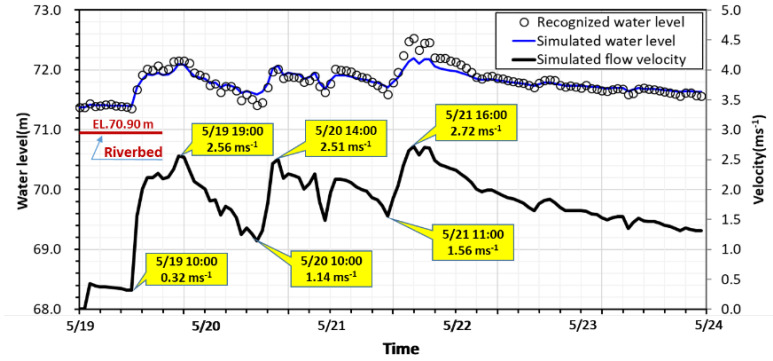
The hydrographs of the recognized hourly water levels and simulated results in the flood event.

**Figure 12 sensors-21-04942-f012:**
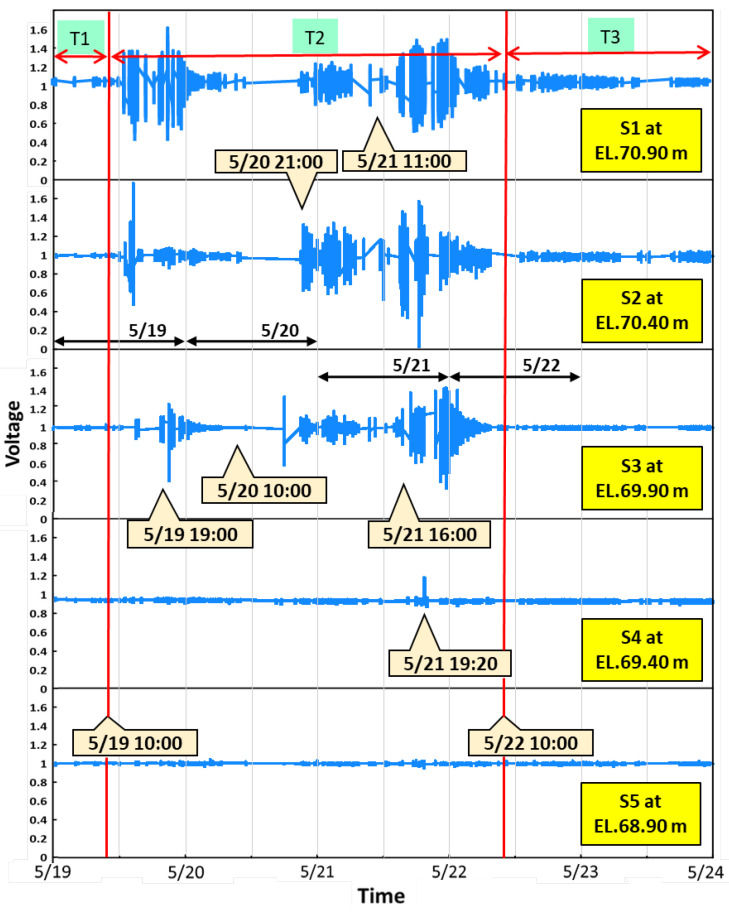
Signal outputs of monitoring sensors during the scour/deposition processes.

**Figure 13 sensors-21-04942-f013:**
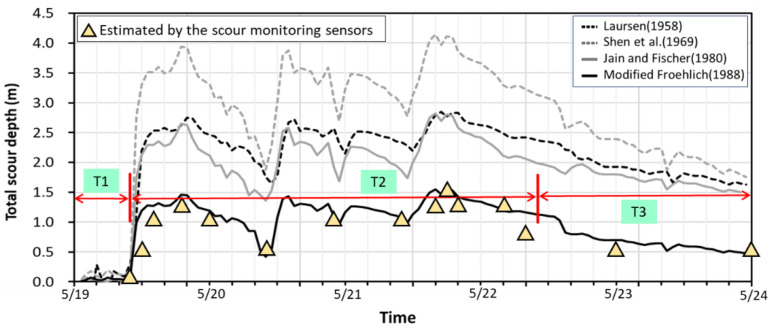
Simulated results of total scour-depth evolution at P28 with various scour formulas.

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
