# Peer review of "The Artificial Intelligence of Things Sensing System of Real-Time Bridge Scour Monitoring for Early Warning during Floods"

_sensors, 2021, doi:10.3390/s21144942_

Round 1

Reviewer 1 Report

This work is focused on the study of developing a scour monitoring system for early warning in highway bridge failure, which includes field monitoring sensors and scour depth simulation. The IoT sensing and wireless communication technology is adopted and a real-time bridge scour monitoring system is proposed and conceptually demonstrated in the laboratory flume.The experimental results show the validation of the system. The method presented in this paper might mean a new contribution to the early warning in highway bridge failure and it is a topic of interest to the researchers in the related areas and a meaningful research for public facilities safety. Also,the English quality of the manuscript is fine and just need minor spell check.

This paper should be accepted and published after minor English spell check.

Page 3,line 78:“……the scour depth is then detected the corresponding signal from the sensor.”. should be “……the scour depth is then detected through the corresponding signal from the sensor.”??

Reviewer 2 Report

A real-time bridge scour monitoring system is proposed and conceptually demonstrated in this study. The monitoring system can measure the scour depth variations during the scour and deposition processes on the riverbed. While the following problems should be clarified.

(1) In this study, the present scour monitoring system is designed with vibration-based arrayed MEMS sensors. MEMS sensors a mature sensing technology, and so what is the innovation of this study?

(2)Please supplement relevant experiments to verify the measurement accuracy of the designed sensor system.

(3) Mask R-CNN deep learning model to estimate water level variations in this study.  Water levels change continuously. While the trained Mask R-CNN is a discrete classifier. How does this discrete classifier estimate continuously changing water levels?

(4) In the process of Mask R-CNN network training, how much data is used in this paper for network training? How to avoid overfitting and underfitting problems during the training process?

(5) Latest references in 3 years should be supplemented.

(6) Figure 12 is not clear. What kind of software was used to draw pictures 11 and 13?

Reviewer 3 Report

The authors treat a topic of utmost importance in structural health monitoring (SHM), that of bridge scouring. To this end, they develop and demonstrate a monitoring system, which is first validated in the laboratory and subsequently applied on a field case study.

Overall, it is a very well-written manuscript. The authors provide analytical details on the monitoring system and its installation. The theoretical background does not create a quantum leap in the field, yet it is one of these cases that really bring the SHM paradigm to the actual engineering practice. As such, the manuscript clearly falls within the aims of the Journal, the quality of the text is good. I propose publication of the manuscript in its current format.

Round 2

Reviewer 2 Report

Indeed, the manuscript is a very well-written manuscript. However, according to the response, the reviewer think that the manuscript lack novelty and The correctness of the results cannot be guaranteed.

(1) According to Response 1, MEMS sensors a mature sensing technology. The study just addresses the sensor packaged in a waterproof stainless steel ball to avoid unnecessary maintenance. 

(2) According to Response 4, the training data set only contains 500 images. For Mask R-CNN, this amount of training data is far from enough to guarantee the correctness of the calculation results.

Author Response

Dear reviewer 2
